# Spatially Resolved Effects of Protein Freeze-Thawing in a Small-Scale Model Using Monoclonal Antibodies

**DOI:** 10.3390/pharmaceutics12040382

**Published:** 2020-04-21

**Authors:** Oliver Spadiut, Thomas Gundinger, Birgit Pittermann, Christoph Slouka

**Affiliations:** 1Research Division Biochemical Engineering, Group for Integrated Bioprocess Development, Institute of Chemical Environmental and Bioscience Engineering, Vienna University of Technology, 1060 Vienna, Austria; oliver.spadiut@tuwien.ac.at (O.S.); thomas.gundinger@tuwien.ac.at (T.G.); 2Head of R&D, ZETA GmbH, Zetaplatz 1, A-8501 Lieboch, 8501 Graz, Austria; Birgit.Pittermann@zeta.com

**Keywords:** monoclonal antibody, freeze-thawing, freezing rate, ice formation rate

## Abstract

Protein freeze-thawing is frequently used to stabilize and store recombinantly produced proteins after different unit operations in upstream and downstream processing. However, freeze-thawing is often accompanied by product damage and, hence, loss of product. Different effects are responsible, including cold denaturation, aggregation effects, which are caused by inhomogeneities in protein concentration, as well as pH and buffer ingredients, especially during the freeze cycle. In this study, we tested a commercially available small-scale protein freezing unit using immunoglobin G (IgG) as monoclonal antibody in a typical formulation buffer containing sodium phosphate, sodium chloride, and Tween 80. Different freezing rates were used respectively, and the product quality was tested in the frozen sample. Spatially resolved tests for protein concentration, pH, conductivity, and aggregation revealed high spatial differences in the frozen sample. Usage of slow freezing rates revealed high inhomogeneities in terms of buffer salt and protein distribution, while fast rates led to far lower spatial differences. These protein and buffer salt inhomogeneities can be reliably monitored using straight forward analytics, like conductivity and photometric total protein concentration measurements, reducing the need for HPLC analytics in screening experiments. Summarizing, fast freezing using steep rates shows promising results concerning homogeneity of the final frozen product and inhibits increased product aggregation.

## 1. Introduction

Antibodies are nowadays widely used for very different purposes in pharmaceutical and biotechnical industries. The importance of monoclonal antibodies (mAB) increased significantly within the last 30 years, starting from the first FDA approved mAB in 1986 (murononab OKT3 for acute organ rejection) from murine mABs to human identical mABs [1,2]. The understanding of the diseases on a molecular level increased the field of application, which can be directly used in activating, inhibiting, or blocking the molecular targets. This led to a massive expansion in the market for mABs. Already in 2013 sales of 75 billion dollars were reached, representing approximately half of the total sales of all biopharmaceutical products with 300 possible new candidates in development in 2015 [2]. The production of mABs is dominantly carried out in mammalian cell cultures, generally using high throughput technologies, design of experiments, and quality by design for increased product titers and high purity of the product in fed-batch or perfusion culture. Titers of above 20 g/L of mABs are reported already, making the downstream and the formulation of the product rich in effort and costs. Details on recent achievements in the upstream development are given in different review articles [3,4,5,6]. Adaption of the downstream to the high titers of the upstream led to a high number of continuous operating systems, which are able to increase operational efficiency and decrease costs especially in chromatography [5,7]. The postulated final aim for mABs processing is a fully integrated continuous process from upstream to downstream [6]. After final formulation, the product needs to be stabilized for transport and storage upon delivery to the final customer. Freeze-thawing is, therefore, an integral step during the manufacturing of monoclonal antibodies. Several benefits are attributed to freezing of the product. The risk for microbial growth is minimized, the product stability is increased, and freezing eliminates agitation and foaming during transport [8,9].

Packaging of the formulated drug is often carried out in polymer containers or polymer/glass syringes. It is reported that phase interfaces, for example between gas and liquid, to the container material or silicon oil residues [10], often affect drug stability [11]. These stability losses are often related to aggregation of the final product. Protein aggregation in pharmaceutical products must be avoided since it usually leads to inactivation of the drug and can even trigger an immunogenic reaction [12]. Furthermore, protein aggregation is nowadays associated with different neural degenerative diseases, like Alzheimer and Parkinson disease [13]. However, aggregation is a rather universal term, describing deviations from the ideal monomeric structure of proteins. Protein aggregates can exhibit different size, shape, and morphology with either covalent or non-covalent bonds [14]. Chemical strategies to reduce aggregation have already been reported. Negative effects of adsorption of the protein to the packaging material can be reduced by adding nonionic substances like Tween 20 or 80 to the product formulation in prefilled syringes [15]. Another strategy to minimize such interaction of the protein to interfaces is freezing of the product and minimizing the contact to other phases [11]. Intended freezing of the product is used for stabilization purposes, like freeze-drying or freeze-thawing, but also happens unintendedly during transport and storage of the formulated product. Freeze-thawing and freeze-drying are used for stabilization and to ensure quality of bioactive drugs, which is the main concern in biopharmaceutical industry [11]. However, different effects are known to favor protein aggregation and/or fragmentation during freezing operations. Cold denaturation is known to destabilize the protein structure upon temperature reduction generally based on reduction of entropy. The detailed understanding of cold denaturation is still under discussion in the community [16,17,18]. Phase separation of high concentration of proteins leads to further problems during freezing. This results in a sole protein phase, which is believed to increase aggregation of the product [19]. Not only protein is highly enriched during freezing but also salts of the buffer system, which can lead to salt crystallization [8,20]. Especially in phosphate-based buffers, this crystallization effect results in high deviations from the set pH upon freezing. All these effects are considered to affect protein stability and promote aggregation of proteins during freezing [8,21,22,23,24,25].

Within this work, we analyzed the effects of different freezing rates on immunoglobin G (IgG) in a phosphate-buffered system using a commercially available small-scale system. While screening experiments on a larger scale are very cost-intensive, especially when using pharmaceutically relevant mABs, small-scale systems decrease time and costs drastically and are, therefore, a perfect tool for determination of freezing characteristics. Different analytics for the process (pH, conductivity) and the product (titer, total protein concentration, and aggregation) were performed in a spatially resolved manner. High effects of the freezing rate on titer and conductivity could be found, while pH exhibited the same value after thawing for all analyzed samples. Total protein concentration and conductivity can be used as easy straight forward analytics to monitor inhomogeneities in buffer and IgG concentration. Aggregation was found to be increased by freezing, however, absolute values stayed below 3% in total. Our results show that high freezing rates are superior to low rates in terms of conformity of the sample and that these results can be transferred to larger-scale systems using the ice formation rate (IFR) as an indication factor.

## 2. Materials and Methods

### 2.1. mABs Production and Purification

Procedures on mABs production using Chinese Hamster Ovary cells (CHOs) in our labs can be found elsewhere [26,27]. The supernatants of different batches containing immunoglobin G (IgG) were pooled and stored at +4 °C. For the purification of the mABs, a self-packed protein A column with KANEKA KanCapA sorbent (Pall Corporation, Port Washington, NY, USA) in combination with an Äkta Pure chromatography system (GE Healthcare, Chigaco, IL, UA) was used. As pre-treatment, the supernatant was filtrated (0.2 µm, Kleenpack) (Pall Cooperation) and diluted (1:2) with binding buffer (50 mM Tris, pH 7.5). The column was equilibrated with binding buffer before the diluted supernatant was loaded with a flow rate of 100–120 cm·h^−1^. After a post-load wash with binding buffer, the column was washed with washing buffer (50 mM Tris, 500 mM NaCl, pH 8.0) before elution was performed with elution buffer (50 mM citrate, 50 mM NaCl, pH 3.0). The IgG was directly eluted into a container filled with binding buffer under slight stirring for neutralization of the pH and to prevent increased IgG aggregation.

The purified antibody was concentrated to a concentration of 25–30 g/L IgG and 10-fold diafiltrated with formulation buffer (20 mM NaH_2_PO_4_, pH 6.0, 9 g/L NaCl, 0.2 g/L polysorbate 80–Tween80) using a cross-flow filtration system (Pall Corporation) with a 30 kD filtration membrane cassette (0.1 m^2^, Pall). The formulated IgG was diluted to a final concentration of 20 g/L with formulation buffer and filtered (0.22 µm, Carl Roth, Karlsruhe, Baden-Würtenberg, Germany) before freezing. The final antibody formulation was analyzed for protein content, IgG content, and by SDS PAGE. The final formulation buffer used in the freezing experiments contained a concentration of 20 g/L IgG.

### 2.2. Freeze-Thawing

The freeze-thawing was performed using a small-scale laboratory freeze-thaw system (LabFreeze system Zeta GmbH, Graz, Austria) with a total volume of 300 mL and a working volume of 200 mL, given in Figure 1a. The cooling supply system was a Huber Unistat 705 (Huber Kältemaschinenbau AG, Offenburg, Germany) with an operating range down to −90 °C and a ZETA control system for applying variable freezing and thawing rates. As freezing fluid, silicon oil 200 Fluid, 5.0 cs (Dow Corning Cooperation, Midland, MI, USA) was used. The systems consisted of 8 temperature probes, 1 to 6 located inside the freezing chamber and probes at the inlet and outlet (compare to Figure 1b). The temperature of all probes was constantly logged during the freezing process.

The formulation buffer for the freezing experiments was 20 mM Na_3_PO_4_, 9 g/L NaCl, and 0.2 g/L Tween 80 at a pH of 6.0 based on results in References [8,28]. We used different freezing rates in order to analyze spatially resolved product quality of the mABs. We tested different rate times in the following experimental approach: The mAB formulation was precooled to +3–+5 °C. Freezing rates were set in four experiments according to Table 1:

The starting temperature of the thermostat was −2 °C and the final temperature was −32 °C in a linear rate with freezing rates given in Table 1. This resulted in different final temperatures among the frozen mAB sample ranging from −15 to −20 °C, varying within the sample block.

The pilot-scale system (PilotFreeze, ZETA GmbH, Graz, Austria) was tested for evaluation of the obtained small-scale effects. It can be used as a scale-up from the small-scale apparatus (Figure 1). It is made of electropolished stainless steel 316L and has a working volume of 700 mL and may be operated down to −90 °C. Its freezing chamber has a cylindrical form (dimensions D and H) that is surrounded by a double jacket for liquid cooling identical to large-scale systems. The system is closed with a stainless steel lid that includes a cooling loop, which allows increasing the cooling area and various operation modes (+/− loop). The lid also contains ports for multiple temperature probes (up to 4) allowing temperature monitoring at variable vertical and horizontal positions. For sampling, a drilling guide is used that ensures positional sampling in frozen state.

### 2.3. Analytics

#### 2.3.1. Sampling

For spatially resolved sampling, we used a fixed drilling machine in a 4 °C cooling room in order to reduce mixing effects during thawing of the frozen block. The sampling points are shown in Figure 2. The drilled cylinders had an inner diameter of 1 cm and the samples were taken through the entire vertical profile. This resulted in 0.98 mL of sample, which was transferred to 1.5 mL Eppendorf-Safe-Lock Tube (Eppendorf, Hamburg, Germany) and de-frozen for subsequent analytics.

#### 2.3.2. Total Protein Concentration

As the final formulation contained no other proteins or peptides except IgG, Bradford method is a well-suited indicator for IgG distribution in the sample block after freezing. The thawed samples were analyzed in terms of total protein concentration. Bradford method was used with standard Bradford reagent at 595 nm measured at a Genesys 20 photometer (Thermo Scientific, Waltham, MA, USA). Samples were incubated for 15 min in adequate concentration to stay within the linear range of Lambert Beer’s law and subsequently measured.

#### 2.3.3. IgG Titer Determination

The content of IgG was determined by Protein A affinity high-performance liquid chromatography (HPLC). Samples were analyzed at 25 °C with a Poros A 20 µm column (2.1 × 30 mm, 0.1 mL, Applied Biosystems) on an Ultimate 3000 HPLC (Thermo) with UV detection. For protection of the column, a pre-column filter (Acquity UPLC col. In-line filter kit, Waters) was used. Ten microliters of sample were injected and the method was performed with a flow rate of 2 mL/h. Used mobile phases were binding buffer A (10 mM NaH_2_PO_4_, 150 mM NaCl, pH 7.5), elution buffer B (10 mM HCl, 150 mM NaCl, pH 2.0), and regeneration buffer C (20% v/v acetic acid). The total run time was 3 min, and UV detection was performed at 280 nm. A calibration curve was performed for a concentration range of 0.5–4.5 g/L using an IgG standard. Samples were diluted to 2.5 g/L in formulation buffer based on total protein content prior to measurement and cooled at 4 °C prior to injection. Dionex Chromeleon software (Thermo Scientific, Waltham, MA, USA) was used for data evaluation.

#### 2.3.4. IgG Aggregate Determination

The content of soluble aggregates was determined by size exclusion high-performance liquid chromatography (SEC-HPLC). Samples were analyzed at 25 °C with a MAbPac SEC-1 column (5 µm 300 Å 4 × 150 mm, Thermo Scientific) on an Ultimate 3000 HPLC (Thermo Scientific) with UV detection. Five microliters of sample were injected and separation was performed with a flow rate of 0.25 mL/h. The mobile phase was 50 mM NaH_2_PO_4_, 300 mM NaCl at pH 6.5. The total run time was 15 min, and UV detection was performed at 210 nm. Samples were diluted to 2.5 g/L in formulation buffer based on total protein content prior to measurement and cooled at 4 °C prior to injection. For calculation of percentage aggregate content, the area under the curve (AUC) of the UV absorbance signal of the aggregate peak was compared with the total AUC of the sample (aggregate and monomer peak). Dionex Chromeleon software (Thermo Scientific) was used for data evaluation.

#### 2.3.5. pH Measurement

The pH value of the samples was measured using a WTW 3110 pH meter (Xylem Inc., Rye Brook, NY, USA) using a micro pH probe.

#### 2.3.6. Conductivity Measurements

In order to get highly accurate data on conductivity, we used impedance spectroscopy as ohmic resistances changes of the media during current flow could be avoided. We pipetted the analyte in a measurement cell exhibiting a classic capacitor design. Impedance measurements were recorded in the range of 10^6^ to 10^0^ Hz with amplitudes between 10 and 100 mV using an Alpha-A High-Resolution Dielectric Analyzer (Novocontrol, Montabaur, Germany) at room temperature. The spectra were evaluated using the ZView software package (Scribner, Southern Pines, NC, USA) and the resistance of the media was extracted and converted into conductivity using the cell geometry in Equation (1):(1)$$cond.\left[\frac{mS}{cm}\right]=1000\times \frac{d\;\left[cm\right]}{area\;\left[cm^{2}\right]\times R\;\left[Ohm\right]}$$
where cond. is the conductivity in S/cm, d is the electrode distance in cm, area is the electrode area in cm^2^, and R is the extracted resistance in Ohm. The values of the samples were always compared to raw buffer values before freezing.

#### 2.3.7. Linear Regression Models

For modeling of interdependencies, we used OriginPro 2016 (OriginLab Corporation, Northampton, MA, USA). Shortly, we fitted the data using linear quick fit routine and extracted the slope and the coefficient of determination (R^2^). The slope is used as quantity of the effect and R^2^ as quality of the effect. All data from the five sampling points were plotted in a contour plot with slope and R^2^ as quantities, x and y as location of the sampling point in the frozen block.

## 3. Results and Discussion

We tested different freezing rates using the small-scale system and analyzed critical product and buffer composition quality. Freezing rates were chosen based on the technical limits of the used experimental system. The lowest possible time was 240 min, while the upper limit was 12 min. The third rate was chosen to reflect a standard freezing process in industry of 120 min (compare to Table 1). The buffer system for the freezing experiments was evaluated through extensive literature research and not alternated during the different experiments. The process was characterized using pH and conductivity and the product was defined in terms of changes of titer and soluble aggregate formation upon freezing. We calculated the ice formation rate (IFR) and used these values to compare the small-scale freezing system to a pilot system with a working volume of 700 mL.

### 3.1. Rate Experiments and Freezing Characteristics

At first, we tested the effectiveness of the freezing chamber and the temperature distribution within the unit. In Figure 3a, the temperature control of the Huber thermostat is presented. Control of the setpoint worked fine within the system, only a delay of some seconds was given within the thermostat. However, at these temperatures, isolation of the system is of utmost importance as heat losses are given through the piping and the freezing chamber itself.

The temperature probes revealed inhomogeneities in the sample chamber during freezing. All probes are visualized in Figure 3b for run #2 exhibiting a freezing rate of 0.25 °C/min. The locations of T1 to 6 are the temperature probes of the lab-scale system, which is given in Figure 1b. The temperature difference between thermostat and freezing chamber with over 10 °C is obvious, showing the importance of proper insulation of the device.

We used these temperature data to calculate the “ice formation rate” (IFR) of each freezing rate in order to get a scalable quantity for different freezing devices. Based on Gibb’s phase rule, temperature can change when only one phase is present (isobar conditions at atmospheric pressure) and volume change occurs within the freezing chamber. The complete phase change is given by the time point of the drop of the last temperature probe in the temperature curves (Figure 3b) and calculated based on Equation (2):(2)$$\mathrm{ice}\;\mathrm{formation}\;\mathrm{rate}=\frac{m}{\mathrm{pc}\times A}$$
where m is the mass inside the freezing chamber in kg, pc is the timepoint of phase change in min, and A is the heat change area in m^2^ (known by computer aided design (CAD) drawings of the apparatus). Results for the time phase change and the ice formation rate are given in Table 2.

We targeted a freezing time of 60 min, theoretically resulting in an IFR of 0.256. However, due to insulation issues, the targeted IFR is only almost reached with the highest freezing rate of 2.5 °C/min (compare to Table 2). However, it must be kept in mind that larger units have higher ice formation rates based on better insulation. What can be clearly dedicated to these results is the very inhomogeneous freezing characteristics, despite clear and reproducible cooling rates with the thermostat. We sketched the freezing characteristics based on our temperature data from all runs in Figure 4. Freezing occurs at first perpendicular to the entrance of the cooling fluid (T2, 6, 3) and expands to the center of the cooling chamber (T4). The last regions, also determining the phase shift and thus the ice formation rate, are the side areas (T1, 5). Based on these effects, the spatial distribution of buffer ingredients and IgG is to be expected.

### 3.2. pH and Conductivity of the Freezing Formulation

As shifts in salt concentration and pH may have severe effects on product stability and activity, we analyzed the pH of the spatially resolved samples and the conductivity as an indicator for redistribution upon freezing of sodium, phosphate, and chloride. Generally, different ions redistribute based on their chemical potential in the solution. This behavior can be explained by different entrapment rates of dissolved bulk components in the growing ice structure, which is determined by the specific solubility and solidification behavior [24].

pH of the buffered system was independent of the freezing rate and location of the sampling after thawing (compare to Appendix A). Maximum changes of 2.7% were found. These changes were within the error of the resistance fit and are, therefore, not significant. The constant pH showed that the buffer capacity of the formulation buffer was high enough, even after redistribution, to compensate for changes in H_3_O^+^ concentration. However, drops in pH during freezing are based on temperature dependence of the dissociation constant (pKa). With phosphate-buffered systems (having pH of 7 at room temperature), pH stays constant until 0 °C and subsequently drops down to pH of 3 at −30 °C, based on crystallization effects [8,20]. After thawing, pH exhibits generally the same value of pH of 7 as before freezing, as seen in the given studies [8,20].

While pH stayed constant after thawing at all tested freezing rates, high changes in conductivity were observed. This was already observed in different other studies [29] but is generally highly dependent on the used buffer system [25]. As salt concentrations were far below 1 M, conductivities are directly proportional to the salt concentrations. Using the molar cross conductivity for the used buffer solution, we received a conductivity of 24.3 mS/cm, which is in good accordance with our results in Table 3 for the initial buffer conductivity. About 80% of conductivity in the solution is based on NaCl and about 20% on Na_3_PO_4._ The changes in conductivity of the different sampling points (shown in Figure 2; Figure 3) are given in Table 3. Highest positive and negative changes are highlighted in gray. The highest salt redistribution was found at low freezing rates of 0.125 °C/min, where concentrations are almost doubled at the side areas of the sampling chamber. The lowest changes were found at the highest tested freezing rate of 2.5 °C/min. The salt concentration was increased by a factor of 0.5. In accordance with temperature data, sensor 1 and 5 faced the highest changes in salt concentrations. While buffer salt concentrations at the initial buffer system were not harmful to the product, far higher concentrations of NaCl, which were present in small regions between the ice crystals, may result in product damage and activity loss during freezing. Differences in final temperature of the freezing chamber were between −15 and −20 °C depending on the location. At temperatures below minus 21.1 °C, we supposedly have traces of liquid water between the ice crystals with high concentrations of salt (over 20% NaCl based on the thermodynamic binary phase diagram water–NaCl), which may cause damage to IgG in these regions. mABs aggregation may be one further consequence, resulting in loss of the valued product. Furthermore, NaCl at lower pH is known to cause significant amounts of soluble aggregates within the downstream process [12].

### 3.3. Effects on Product Quality

We analyzed the IgG product titer and the aggregation of IgG during freezing. Both values were measured using extensive HPLC analytics, which is cost- and time-intensive. Therefore, we also checked straight forwards protein analytics, like Bradford, in order to reduce the analytical complexity. mAB solutions in water (with different buffer systems) have a phase separation at temperatures of about −6 to −7 °C at our respective concentrations, which resulted in a separate pure mAB phase in this temperature range [19]. Results for the titer analytics are given in Table 4. The highest changes in protein concentration were found within the sample regions S2 and S5, which is in accordance with the general freezing characteristics in Figure 4. These highly enriched sampling areas may—as discussed before—be the hot spot for protein aggregation as concentrations in solution are even increased compared to the bulk. As protein aggregation is believed to be the major effect for inactivation of the biological activity [23], we analyzed the aggregation as critical quality attributes (CQA) during the freezing.

Soluble agglomeration (in fact dimerization or trimerization) was enhanced by concentrating the protein and increasing salt concentration [28]. Both effects were present in the whole sample but highly located at the side regions, where these effects are believed to be drastically increased. In the base material, we found an agglomeration content of about 2% on average for all tested runs, being a result of storage at +4 °C of the formulation buffer prior to the experiments. Within the freezing experiments, the highest change in agglomeration was found at low and intermediate freezing rates with 50.3% enrichment in agglomerates (details given in Appendix A). This resulted in an agglomerate amount of 2.6% showing very low aggregation compared to other studies [23]. However, in this study, differences in pH and KCl were considered, resulting in very different amounts of aggregation based on these variables. In general, high amounts of salt are believed to have a high effect on the aggregation behavior of immunoglobins [12,30]. Summarizing, longer freezing duration and several freezing cycles may enhance agglomeration severely based on the found inhomogeneities within the system.

In this study, we were able to reveal spatial effects of freeze-thawing in a small-scale model. We investigated different applicable freezing rates and introduced the ice formation rate (IFR). The lowest changes in conductivity and protein redistribution were found at highest freezing rates with an IFR of 0.22 kg/(min·m^2^) and are regarded as favorable for freeze-thawing procedures. However, as these freezing rates are the lower limit in large scale industrial systems, we used the IFR evaluation to compare our results to an industrial system. To ease product analytics, we tried to get a mechanistic description of the used measured quantities to find suitable correlations.

### 3.4. Multivariate and Mechanistical Description of mAB Freezing

Freezing within the sample block was shown to occur in an inhomogeneous manner with ice nucleation and freezing starting at the areas perpendicular to the in/outlet of the cooling fluid. The latest point of freezing happened in the side regions of the sample block indicated by the temperature profiles. This resulted in high accumulation of salts in the formulation buffer as well as the target protein. In a first approach, we analyzed the datasets (precentral change to the start material) for IgG titer, conductivity, and mAB aggregation measured at 210 nm at the UV detector of the HPLC (see Appendix A) using principal component analysis (PCA) with the software package for multivariate analysis in OriginLab2016 (OriginLab Corporation). We analyzed the correlation matrix of the data set and two principal components could describe the variance in the data set of the six samples to 92.2% cumulative. The score plot for the PCA is given in Figure 5.

Using PCA we can clearly distinguish between effects of freezing in three distinct regions. Samples 6, 4, and 1 are located at the earliest points of freeze next to the inlet of the cooling fluid (compare to sketch in Figure 4). Sample 3 is located right in the middle of the block and sample 2 and 5 are at the side regions, respectively. Principle component 1 reflects, therefore, mainly the location of the taken samples within the frozen block. Details on eigenvectors are given in the Appendix A. Principle component 2 is mainly affected by conductivity and mAB aggregation values of run #4. Therefore, the freezing rate seems to mainly dominate principal component 2, as the lowest changes are always found within run #4.

As HPLC measurements are laborious and cost-intensive compared to techniques like conductivity measurement and photometric assays like Bradford total protein measurements, we investigated the interdependencies between protein titer measured by HPLC with Bradford and conductivity measurements. The first correlation is done between conductivity and HPLC titer using linear regression as can be seen in Figure 6a. We plotted the change in % of conductivity vs. the change in % of titer and fitted with a linear regression using OriginLab2016. We extracted the slope of the linear regression as an indicator for strength of the effect and R^2^ as a quality indicator. Values for an IFR of 0.192 kg/(min*m^2^) are averaged values of run #1 and 2.

In Figure 6b,c the results for slope and the R^2^ (coefficient of determination) are shown. Fit quality is good enough in the region of interest, namely the center (Sample 3) and the side regions (Sample 2 and 5). Shallow slopes indicate low change of titer, while high change of conductivity is observed. Steep slopes indicate inversed behavior. So, slopes of the linear regression of about 1%/% change are optimal since conductivity data can be directly used to measure the titer concentration within the sample (compare to green region in Figure 6b). Conductivity data can be used as an indicator of inhomogeneities in IgG concentration. Conductivity measurements might be directly applied during the freezing process as inline probes for direct detection of critical concentrations of IgG. When applying conductivity in the frozen state, pH changes due to different temperatures in the solid ice bulk have to be taken into account. These changes are a result of the temperature dependence of the pH value and phosphate crystallization [24], which may have also effects on the titer/conductivity relation. Furthermore, applications of surface microelectrodes in the solid ice state may be reasonable to get high local information about protein concentrations and especially enrichment hot spots as shown in literature [21,31]. Bradford results are very similar to the HPLC titer. Slight changes in effect strength are visible in Figure 6c. The slope of the linear regression is the highest at the side regions, while centered areas are lower. R^2^, given in Figure 6e, is reasonable in the centered and side regions, while in the areas of first freezing (Samples 6, 4, and 1)—where we did not see much effects of concentration changes—linear regression is poor. Total protein concentration by Bradford is an adequate tool to measure effects on the concentration redistribution in freezing experiments and will be primarily used in the following evaluation.

### 3.5. IFR as Key Parameter for Freeze/Thawing Experiments

Conductivity and Bradford measurements are a well-suited indicator for protein redistribution in the scale down system. Therefore, we used conductivity and total protein concentration as tools to analyze freezing characteristics in a pilot-scale with 700 mL volume, but complex three-dimensional geometry. In Figure 7a, the representative sampling is given, with three samples from the top (horizontal) and three samples next to the cylinder jacket (vertical). In sum, six samples were thawed and analyzed subsequently using straight forward conductivity and Bradford measurements as done in the lab-scale experimental approach.

For effect quantification, we took the maximal positive changes of conductivity and total protein concentration measured with Bradford. We also found high values for salt and protein depletion in the six samples, but as enrichment is believed to cause negative effects on protein stability we focused on the highest positive values. The highest changes within all tested runs from small-scale and pilot-scale are summed up in Figure 7b, resulting in similar behavior of small-scale and pilot-scale for conductivity and total protein concentration. Small-scale values for IFR are higher than the counterparts for the pilot-scale system. Based on these results, fast freezing rates (higher IFR) result in lower maximal changes in the tested samples. High freezing rates have beneficial effects in the tested systems, however, IFR can only be used as key parameter within a given geometry, as offsets of the linear correlations are highly diverse (Figure 7b). In fact, for both geometries, high freezing rates resulted in decreased agglomeration of salts and protein within the analyzed samples. A correlation factor of 1.58 was calculated and can be used to transfer results from the small-scale system to the pilot-scale. This factor has to be calculated for every used system as empirical value based on differences in freezing characteristics depending on the difference in geometry. While heat transfer in the small-scale is only done by the long side areas, the heat transfer pilot-scale is given through the whole mantle and loops inside the core. One drawback of high freezing rates seems to be absolute loss of protein within the pilot-scale test. In general, the added concentration of Tween 20 is sufficient for stabilization in the lab-scale experiments as Tween is known in literature to prevent insoluble aggregate formation [32]. The total protein concentration, after thawing of the 700 mL, is analyzed and found to be reduced by 28%. This might be the result of insoluble aggregate formation (higher polymerization grade) within the tested run being a result of different discussed effects [33]. The lower IFR may be a reason for this high loss of protein on the pilot-scale, which Tween 20 is not able to counter-stabilize in the used formulation. SEC is, furthermore, not capable of detecting the high particle size of insoluble aggregates. For a detailed analysis on effects of high IFR, different analytic techniques, like microflow imaging of insoluble particles, may be used to get further insights [34].

## 4. Conclusions

In this study, we tested the effects on the unit operation freeze-thawing onto an industrial relevant monoclonal antibody in a small-scale model. As costs for screening experiments are very high in pilot systems or even industrially used systems, we wanted to test if small-scale systems can perform as an adequate model platform for high volume systems.

Therefore, a small-scale freeze-thaw system was used to screen for spatial resolution of different quantities. Process and product quantities were analyzed within the experiments using this setup. The spatial resolution of protein quality was evaluated, and it was shown that side regions of the freezing device show the highest concentrations of product after the freeze-thaw circle. Total protein concentration and conductivity measurements gave highly accurate information about the spatially resolved protein concentration without need to measure time and cost consuming HPLC methods. pH was unaffected upon salt concentration changes after the freeze circle. Soluble aggregates were increased during processing especially with low freezing rates. However, absolute values were in our study far below 3% of the total area of IgG analyzed with SEC HPLC. High freezing rates, expressed in the IFR, resulted in the lowest changes of all measured quantities, and are, therefore, highly recommended for industrial applications.

The IFR trends were tested on a large-scale system in evaluation of conductivity and total protein measurements. Despite the higher complexity of the system, the trends from the small-scale system supported the experiments performed in the large-scale system. The ice formation rate (IFR) could be used as a key parameter for freezing experiments, with slight adaptions based on the used geometry. Higher IFR increases the quality of the product not only for the scale down system but also on a larger scale. Therefore, small-scale freezing devices are highly supportive in gaining insight into freezing trends with far lower need in volumes and respective cost for the experiment and enable screening experimental design, which could not be afforded in larger scale.

## Figures and Tables

**Figure 1 pharmaceutics-12-00382-f001:**
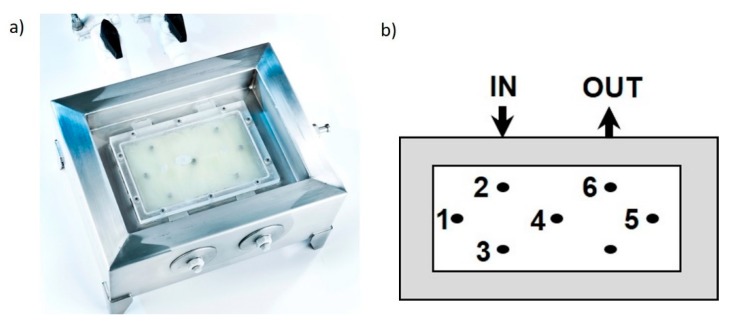
(**a**) Actual freezing chamber of the freezing unit with respective temperature probes inside the chamber; (**b**) sketch of the chamber and location of temperature probes.

**Figure 2 pharmaceutics-12-00382-f002:**
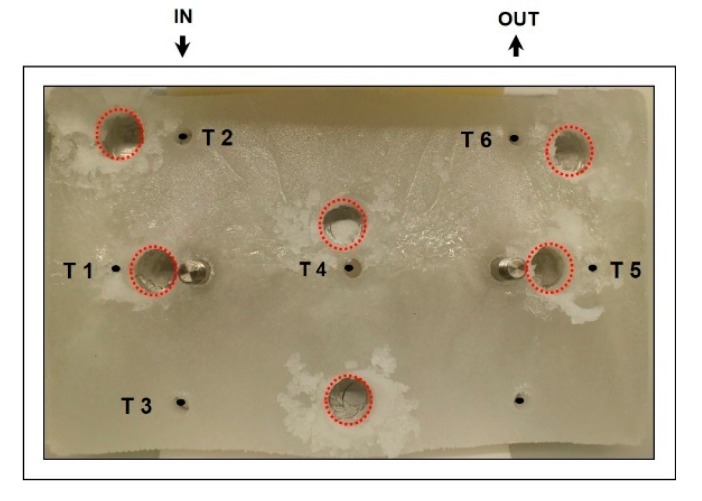
Frozen sample block after freezing cycle. The dots represent the temperature probes (compare to Figure 1b) including legend T 1 to T 6 including temperature measurement at the inlet and the outlet (IN and OUT). Red dotted circles represent the drilling holes for spatially resolved sampling.

**Figure 3 pharmaceutics-12-00382-f003:**
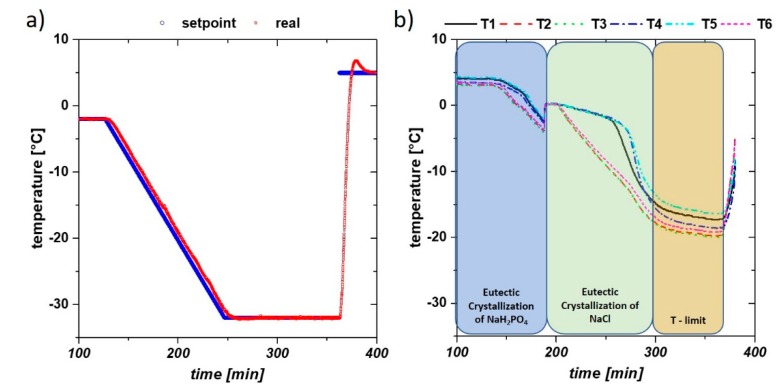
(**a**) Thermostat setpoint and measured temperature at freezing rate of 0.25 °C/min; (**b**) actual temperature of the thermocouples inside the chamber (sketched in Figure 1b), showing inhomogeneous freezing in the chamber. The final temperature had a difference of over 10 °C between thermostat and chamber marked as T limit of the cell.

**Figure 4 pharmaceutics-12-00382-f004:**
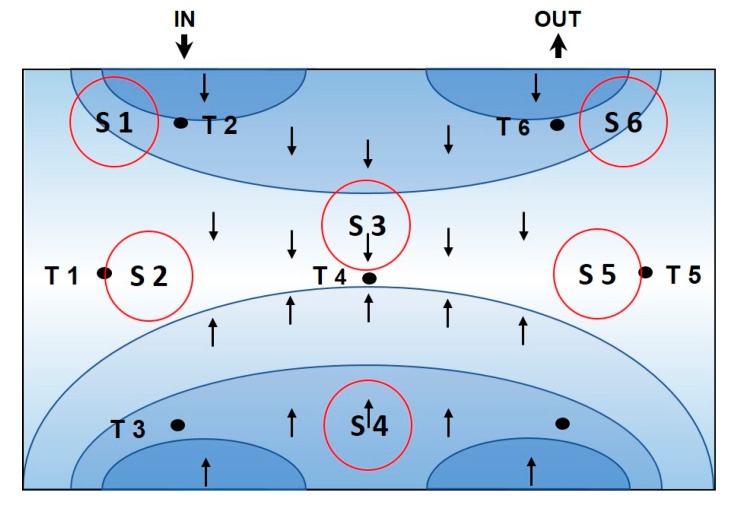
Sketch of the freezing characteristics inside the small-scale device based on the temperature data. Red circles indicate the points of sampling, T the temperature probes. Blue ellipsoids and arrows indicate the direction of the freezing front.

**Figure 5 pharmaceutics-12-00382-f005:**
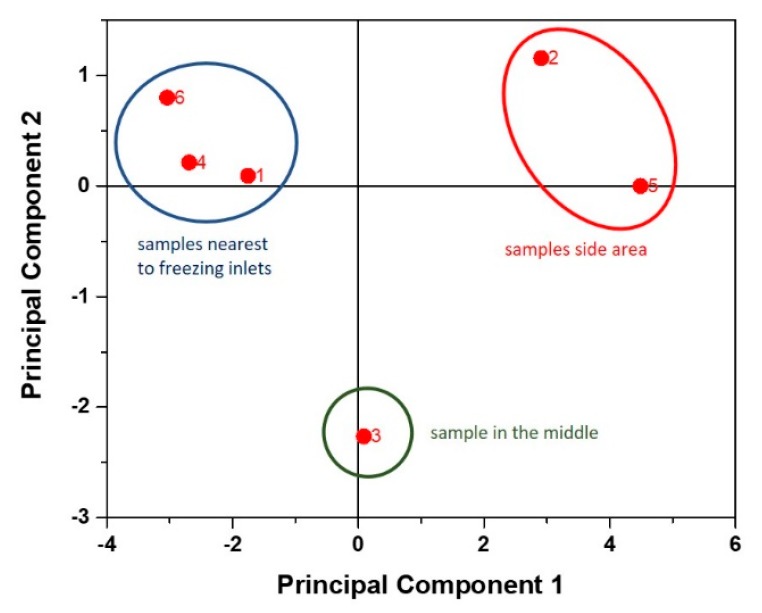
Principal component analysis (PCA) of data sets for titer measurement, conductivity, and monoclonal antibodies (mAB) aggregation. Principle component 2 is mainly driven by titer and mAB aggregation data of run #4 exhibiting the highest ice formation rate (IFR) rate for the lab-scale experiment.

**Figure 6 pharmaceutics-12-00382-f006:**
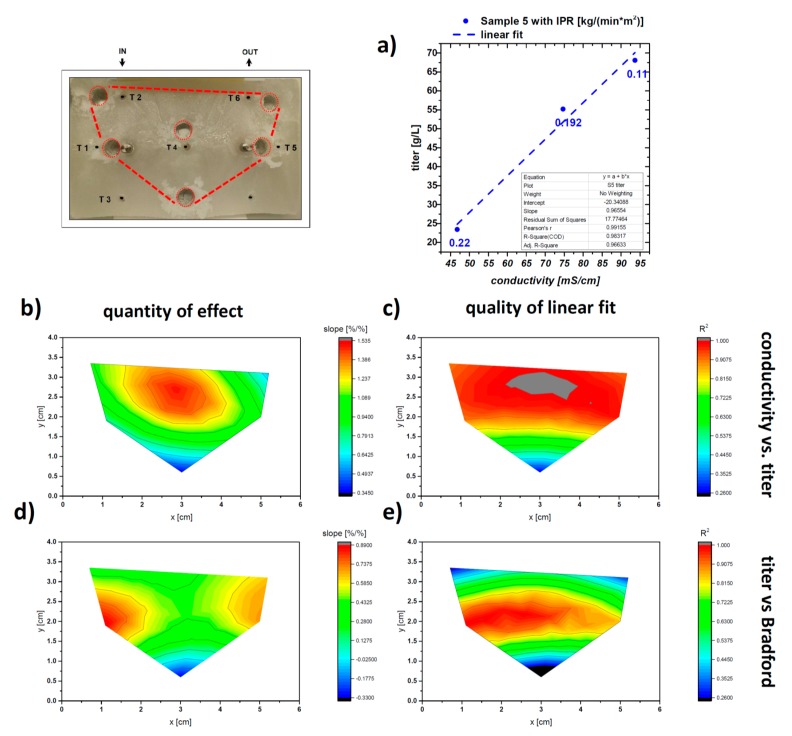
Analysis of the samples after freezing (**a**) exemplary linear fit of conductivity vs. titer; (**b**) Correlation of conductivity vs. titer, evaluated via linear regression based on the sampling points in Figure 2. A high correlation between conductivity and mAB titer is found with a R^2^ (**c**) of 0.8 to 1 in the upper part of the sampling area: (**d**) Titer correlated to Bradford total protein measurement. In the regions of interests, a well-suited correlation can be found (**e**). HPLC measurements could be reproducibly replaced by photometric Bradford assays.

**Figure 7 pharmaceutics-12-00382-f007:**
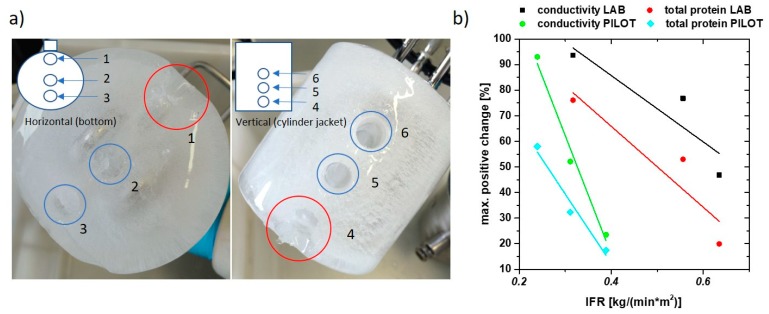
(**a**) Sampling scheme for tests with the pilot-scale system. Three samples were taken at the bottom of the frozen sample (horizontal) and three samples next to the jacket in different height (vertical); (**b**) conductivity and total protein concentration change in % vs. IFR. Very similar dependence for small scale and pilot scale is found.

**Table 1 pharmaceutics-12-00382-t001:** Freezing setpoints of the Huber thermostat for given experiments.

Run	Time	Freezing Rate (HUBER)
#1	120 min	0.25 °C/min
#2	120 min	0.25 °C/min
#3	240 min	0.125 °C/min
#4	NO rate	~2.5 °C/min

**Table 2 pharmaceutics-12-00382-t002:** Ice formation rate is calculated using obtained times of phase change by the “last point of freeze”, which determines time of phase change.

Small-Scale	Unit	Target	Run #1	Run #2	Run #3	Run #4
Freezer content	m [kg]	0.2				
Time-phase change	t [min]	60	80	80	140	70
Heat exchange area	A [m^2^]	0.013				
Ice formation rate	[kg/(min∗m^2^)]	0.256	0.192	0.192	0.110	0.220

**Table 3 pharmaceutics-12-00382-t003:** Changes in conductivity measured by impedance spectroscopy. Higher conductivity reflects higher number of ions in the liquid sample. Gray marking highlights the most prominent changes in conductivity.

Sample	#1-120 min	Change	#2-120 min	Change	#3-240 min	Change	#4-NO Rate 12 min	Change
	[mS/cm]	[%]	[mS/cm]	[%]	[mS/cm]	[%]	[mS/cm]	[%]
Start material	21,06	-	20,21	-	19,95	-	19,36	-
Sample 1	21,20	0,6	20,25	0,2	17,16	−14,0	19,77	2,1
Sample 2	34,26	62,7	33,57	66,1	36,82	84,6	27,48	42,0
Sample 3	24,97	18,6	23,04	14,0	24,31	21,9	25,05	29,4
Sample 4	22,53	7,0	19,77	−2,1	17,99	−9,8	17,71	−8,5
Sample 5	39,50	87,5	32,71	61,9	38,64	93,7	28,41	46,8
Sample 6	18,12	−14,0	20,50	1,5	16,17	−18,9	18,72	−3,3

**Table 4 pharmaceutics-12-00382-t004:** Changes in protein titer as CQA measured at 280 nm. Highest changes in titer are again found within the side areas of the freezing chamber. Gray marking highlights the most prominent changes in titer.

Sample	#1-120 min	Change	#2-120 min	Change	#3-240 min	Change	#4-NO Rate 12 min	Change
	[g/L]	[%]	[g/L]	[%]	[g/L]	[%]	[g/L]	[%]
Start material	25,11	-	23,97	-	23,93	-	23,6	-
Sample 1	25,31	0,8	24,46	2,0	21,41	−10,5	25,75	9,1
Sample 2	35,3	40,6	35,78	49,3	36,06	50,7	27,85	18,0
Sample 3	25,49	1,5	26,96	12,5	27,67	15,6	30,01	27,2
Sample 4	24,94	−0,7	24,8	3,5	22,47	−6,1	24,11	2,2
Sample 5	42,65	69,9	33,69	40,6	40,22	68,1	29,13	23,4
Sample 6	24,16	−3,8	22,96	−4,2	21,69	−9,4	23,71	0,5

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
