# Peer review of "Spatially Resolved Effects of Protein Freeze-Thawing in a Small-Scale Model Using Monoclonal Antibodies"

_pharmaceutics, 2020, doi:10.3390/pharmaceutics12040382_

Round 1

Reviewer 1 Report

Significant review for the accuracy of the units used in the document needs to be performed by the authors as several errors appear to be evident.  example line 180: flow rate of 0.25 mL/h would not be sufficient to elute anything off of the column in 15 min.

The description of buffer system used appears to not be listed correctly and is not consistent with the pH values listed: line 122: Na3PO4 at pH 6.0; line 170: 10 mM NaH2PO4 at pH 7.5; line 180/181:  NaH2PO4 at pH 6.5;  line 281 Na3PO4.

Line 36:  Date of "1886" is inaccurate

Line 51:  word "costumer" is most likely a typo.  Does the author mean "customer"?

Line 290/291:  the "-" should be kept on the same line as the number

Figure 5: poor quality of graphics such that I cannot read parameters on line plot.

Line 374:  word :"too" is a typo

Author Response

Significant review for the accuracy of the units used in the document needs to be performed by the authors as several errors appear to be evident. example line 180: flow rate of 0.25 mL/h would not be sufficient to elute anything off of the column in 15 min.

The description of buffer system used appears to not be listed correctly and is not consistent with the pH values listed: line 122: Na3PO4 at pH 6.0; line 170: 10 mM NaH2PO4 at pH 7.5; line 180/181:  NaH2PO4 at pH 6.5;  line 281 Na3PO4.

I think the reviewer mixed up the buffer system with the HPLC buffers. The buffer in line 122 is the one used for the freezing experiments, while all other buffers are HPLC running buffers and are used for analytics.

Line 36:  Date of "1886" is inaccurate

Changed accordingly

Line 51:  word "costumer" is most likely a typo.  Does the author mean "customer"?

Yes, the reviewer is completely right, we changed that accordingly

Line 290/291:  the "-" should be kept on the same line as the number

We changed the sign to the word “minus”

Figure 5: poor quality of graphics such that I cannot read parameters on line plot.

Increased quality of the figure

Line 374:  word :"too" is a typo

Changed accordingly

Reviewer 2 Report

Manuscript deals with interesting topic, however contains many bad descriptions and explanations. Title is confusing. Maybe i am wrong, but i would like authors what can reader expect from different part od the title.

In Abstract authors  are writing abount properties measured in frozen state. It it possible to measure pH in in ice? In my opinion they assumed about the frozen sample propertieswithin the vessel after thawing. But this is only my guess.  

I have many other suggestions and questions regarding the content

  1. »thermodynamic cold denaturation« is in my opininon wrong expression
  2. « buffer active ingredients« is in my opininon also wrong expression
  3. QbD apprevation should be avoided as it is used only once in the text.
  4. I think that expression »freezing ramp« ir very rarely used in scientific literature.
  5. Line 61 absorption should in my opinion replaced by adsorption.
  6. Line 62 Tween in nonionic substanece not monoionic.
  7. In the Introduction chapter only aim should be described. Reason of freze-thaw process use in the industry of biologicals should be explained. As i am not an expert in the field i would expect that only freezing contributes to improved stability of the water protein dispersion.
  8. Line 95: Abbrevation CHO should be explained.
  9. Line 135 Pilot freeze is listed for the second time here. It thould be avoided.
  10. Line 211 Did you want to test or you tested the effectiveness of the freezing chamber and the temperature...? Authors should be clear in the explanations.
  11. Colors on the Fig 3 should be designated.
  12. Line 230. Most probably 3b should be wtitten here instead of 3a.
  13. Line 239. It woud be convenient to explain how pc and A were determined?

Line 247. T is not clear what means »The targeted IFR is almost reached with the highest freezing rate of 2.5°C/min«?

  1. Blue color and arrows should be explained on Fig. 4.
  2. Line 269. I think that it is neccessary to explain what means redistribution somewhere in the text?
  3. Line 259. No results are given on pH values while discussion exists.

17 Line 329. Why authors use expression »spatial effects« of freeze-thawing?  Here a kind of conclusion is made which is strange. Authors  should write somewhere in the text which results are expected to expect apropriate stability of the product.

I stopped reding the text here as it gets more and more confusing, giving no meaninful data i would be interested in. Authors tried to find some linear correlations without explaining why. Multivariate analysis would be more appropriate. English is poor and the text is badly structured. It seems that authors have little experience in writing scientific articles. They should involve someone who does and restructure what they did and then submit the manuscript.

Author Response

Manuscript deals with interesting topic, however contains many bad descriptions and explanations. Title is confusing. Maybe i am wrong, but i would like authors what can reader expect from different part of the title.

In Abstract authors are writing about properties measured in frozen state. It is possible to measure pH in in ice? In my opinion they assumed about the frozen sample properties within the vessel after thawing. But this is only my guess.  

Properties in the solid phase are hardly accessible, especially in isolated freezing devices, therefore, as also described in chapter “2.3.1 Sampling”, the samples were taken in the frozen state and subsequently analyzed after thawing. Based on the results we tried to understand effects upon freezing

I have many other suggestions and questions regarding the content

»thermodynamic cold denaturation« is in my opinion wrong expression

Changed to only cold denaturation

« buffer active ingredients« is in my opinion also wrong expression

Changed to buffer ingredients

QbD apprevation should be avoided as it is used only once in the text.

Changed accordingly

I think that expression »freezing ramp« is very rarely used in scientific literature.

Changed all ramp to rate

Line 61 absorption should in my opinion replaced by adsorption.

Changed accordingly

Line 62 Tween in nonionic substance not monoionic.

Changed accordingly

In the Introduction chapter only aim should be described. Reason of freeze-thaw process used in the industry of biologicals should be explained. As I am not an expert in the field I would expect that only freezing contributes to improved stability of the water protein dispersion.

We added a part in the introduction with benefits of freeze thawing during mAB processing:

Freeze thawing is therefore an integral step during the manufacturing of monoclonal antibodies. Several benefits are attributed to freeing of the product. The risk for microbial growth is minimized, the product stability is increased and freezing eliminates agitation and foaming during transport [8,9]

Line 95: Abbreviation CHO should be explained.

Added Chinese Hamster Ovary cells

Line 135 Pilot freeze is listed for the second time here. It should be avoided.

We think the reviewer meant LabFreeze, we changed it accordingly

Line 211 Did you want to test or you tested the effectiveness of the freezing chamber and the temperature...? Authors should be clear in the explanations.

Changed to tested.

Colors on the Fig 3 should be designated.

We modified Figure 3 in order to make it suitable for black and white print

Line 230. Most probably 3b should be written here instead of 3a.

Actually the text states Figure 1 b and should show the location of the temperature probes in the sketch, to compare for the temperature behavior.

Line 239. It would be convenient to explain how pc and A were determined?

Pc is described in the text, but we tried to clarify this point further:

The complete phase change is given by the timepoint of the drop of the last temperature probe in the temperature curves (Figure 3 b) and calculated based on Eq. 2:

The heat exchange area was calculated through the CAD drawings of the freezing system.

Line 247. T is not clear what means »The targeted IFR is almost reached with the highest freezing rate of 2.5°C/min«?

We added this part to make it easier for the reader to follow:

We targeted a freezing time of 60 min, theoretically resulting in a IFR of 0.256. However due to insulation issues, the targeted IFR is only almost reached with the highest freezing rate of 2.5°C/min (compare to Table 2).

Blue color and arrows should be explained on Fig. 4.

Added explanation in the caption and modified the arrows.

Line 269. I think that it is necessary to explain what means redistribution somewhere in the text?

We added: redistribution upon freezing of sodium, phosphate and chloride.

Line 259. No results are given on pH values while discussion exists.

That it correct. We wanted to highlight that changes in pH are negligible in the samples. We will put the detailed analysis in the Supplementary part.

17 Line 329. Why authors use expression »spatial effects« of freeze-thawing? Here a kind of conclusion is made which is strange. Authors should write somewhere in the text which results are expected to expect appropriate stability of the product.

The sampling was spatially resolved and as the results also suggest, we got a high difference in conductivity, titer etc. within the sampling block. Spatial sampling of the frozen species is not performed in a larger scale system as it is very cost intensive and often is even not possible from an experimental point of view (cooling coils inside the ice block). Freeze-thawing is the respective unit operation; therefore, this wording is used.

I stopped reading the text here as it gets more and more confusing, giving no meaningful data I would be interested in. Authors tried to find some linear correlations without explaining why. Multivariate analysis would be more appropriate. English is poor and the text is badly structured. It seems that authors have little experience in writing scientific articles. They should involve someone who does and restructure what they did and then submit the manuscript.

We took the criticism of the reviewer very serious. Therefore, we performed a principle component analysis of the analyzed quantities, like titer, conductivity, etc. The results support the hypothesis drawn in the previous chapter. The location of the sample within the frozen block is very important, as well as the freezing rate. We added Figure 5 and following part:

In a first approach we analyzed the datasets (precentral change to the start material) for IgG titer, conductivity and mAB aggregation measured at 210 nm at the UV detector of the HPLC (see Supplementary part) using principle component analysis (PCA) with the software package for multivariate analysis in OriginLab2016 (OriginLab, MA, USA). We analyzed the correlation matrix of the data set and two principle components could describe the variance in the data set of the six samples to 92.2 % cumulative. The score plot for the PCA is given in Figure 5.

Using PCA we can clearly distinguish between effects of freezing in three distinct regions. Samples 6, 4 and 1 are located at the earliest points of freeze next to the inlet of the cooling fluid (compare to sketch in Figure 4). Sample 3 is located right in the middle of the block and sample 2 and 5 are at the side regions respectively. Principle component 1 reflects therefore mainly the location of the taken samples within the frozen block. Details on eigenvectors are given in the Supplementary Part. Principle component 2 is mainly effected by conductivity and mAB aggregation values of run #4. Therefore, the freezing rate seems to mainly dominate principle component 2, as the lowest changes are always found within run #4.

Furthermore, we tried to clarify the subsequent analysis using the linear regression model. Our aim in this part was too reduce the need for analysis, especially by HPLC, and use easy to measure quantities like conductivity and Bradford. We adapted the part in the following way to make it easier for the reader to follow:

As HPLC measurements are laborious and cost intensive compared to techniques like conductivity measurement and photometric assays like Bradford total protein measurements, we investigated the interdependencies between protein titer measured by HPLC with Bradford and conductivity measurements. The first correlation is done between conductivity and HPLC titer using linear regression as can be seen in Figure 6 a. We plotted the change in % of conductivity vs. the change in % of titer and fitted with a linear regression using OriginLab2016. We extracted the slope of the linear regression as indicator for strength of the effect and R2 as quality indicator. Values for an IFR of 0.192 kg/(min*m2) are averaged values of run # 1 and 2.

In Figure 6 b and c the results for slope and the R2 (coefficient of determination) are shown. Fit quality is good enough in the region of interest, namely the center (sample 3) and the side regions (sample 2 and 5). Shallow slopes indicate low change of titer, while high change of conductivity is observed. Steep slopes indicate the inversed behavior. So, slopes of the linear regression of about 1 %/% change are optimal, since conductivity data can be directly used to measure the titer concentration within the sample (compare to green region in Figure 6 b). Conductivity data can be used as indicator for inhomogeneities in IgG concentration. Conductivity measurements might be directly applied during the freezing process as inline probes for direct detection of critical concentrations of IgG. When applying conductivity in the frozen state pH changes due to different temperatures in the solid ice bulk have be taken into account. These changes are a result of the temperature dependence of the pH value and phosphate crystallization [24], may have also effects on the titer/conductivity relation

Reviewer 3 Report

The paper entitled “Spatially resolved effects of protein freeze-thawing in a small-scale model using monoclonal antibodies” by Spadiut et al described spatially diverged ice formation profiles of the laboratory scale freezing container for the protein stock solution. The apparatus was specially designed for the pilot experiment of the industrial scale protein freezing-thaw process. The authors systematically collected the data points and found large inhomogeneity in terms of either the salt concentration or the protein concentration. Several correlations between protein concentration, amount of aggregation, and remained protein activity against conductivity and pH were mathematically analyzed. Since the conditions for freezing protein solution as storage, especially therapeutic antibodies, are extensively studied in industry, there are only limited information published in literature. In this context, this manuscript may scientifically sound. This reviewer likes to point out some issues to be addressed below.

Minor points

1     At first, probably because of the editorial system, this reviewer could not find the supplementary figures and supplementary tables at the reviewing process. This reviewer likes to criticize with them in the next reviewing process for the revised version of this manuscript carefully.

2     This reviewer has to main comments to make: It would certainly strengthen the appeal of the manuscript to the journal Pharmaceutics’s readership in the field if the authors will explain in the introduction section or the very beginning part of the results section why the four experiments of freezing profile were designed? While many peripheral correlations were well discussed, this reviewer felt uneasy whether the most important part of the experimental design was finally achieved or not. Thus, the reviewer recommends to strengthen the point.

3     In the figure 3 legends. There are nor explanation of the colors of the graph lines. Please clarify the line colors with their corresponding temperature probes in Figure 1b.

4     In the figure 4 legends. There are no explanation of the four different grade of the shaded colors in the figure (from white to bluish). In addition, there is no explanation of the arrows indicated in the figure.

5     This is the reviewer’s personal interest. Although the operation manual of the LabFreeze apparatus may not allow, how about the alternating direction of the freezing fluid silicon oil with a switching bulb for a short period? Is the observed inhomogeneity of the cooling efficiency related to the flow direction of the freezing fluid?

Author Response

The paper entitled “Spatially resolved effects of protein freeze-thawing in a small-scale model using monoclonal antibodies” by Spadiut et al described spatially diverged ice formation profiles of the laboratory scale freezing container for the protein stock solution. The apparatus was specially designed for the pilot experiment of the industrial scale protein freezing-thaw process. The authors systematically collected the data points and found large inhomogeneity in terms of either the salt concentration or the protein concentration. Several correlations between protein concentration, amount of aggregation, and remained protein activity against conductivity and pH were mathematically analyzed. Since the conditions for freezing protein solution as storage, especially therapeutic antibodies, are extensively studied in industry, there are only limited information published in literature. In this context, this manuscript may scientifically sound. This reviewer likes to point out some issues to be addressed below.

Minor points

1     At first, probably because of the editorial system, this reviewer could not find the supplementary figures and supplementary tables at the reviewing process. This reviewer likes to criticize with them in the next reviewing process for the revised version of this manuscript carefully.

We will add the supplementary material adequately.

2     This reviewer has to main comments to make: It would certainly strengthen the appeal of the manuscript to the journal Pharmaceutics’s readership in the field if the authors will explain in the introduction section or the very beginning part of the results section why the four experiments of freezing profile were designed? While many peripheral correlations were well discussed, this reviewer felt uneasy whether the most important part of the experimental design was finally achieved or not. Thus, the reviewer recommends to strengthen the point.

We alternated the freezing rate of the system as it was the only process variable we were capable of alternating. The geometry is fixed within the tested systems. Introducing also variations in the buffer system would have put us to different problems: first using even a fractional factorial design (with buffer components, and freezing rates) the number of freezing experiments would have increased drastically. We had only a single device for the experiments and a limited amount of mAB, which were exclusively cleaned by the authors for this study. Therefore, our approach was to keep buffer constant and rather concentrate on spatial resolved sampling with alternating the freezing rates. We added following part:

We tested different freezing rates using the small-scale system and analyzed critical product and buffer composition quality. Freezing rates were chosen based on the technical limits of the used experimental system. The lowest possible time was 240 min, while the upper limit was 12 min. The third rate was chosen to reflect a standard freezing process in industry of 120 min (compare to Table 1). The buffer system for the freezing experiments was evaluated through extensive literature research and not alternated during the different experiments.

3     In the figure 3 legends. There is no explanation of the colors of the graph lines. Please clarify the line colors with their corresponding temperature probes in Figure 1b.

We clarified this picture and tried to make it better readable

4     In the figure 4 legends. There is no explanation of the four different grade of the shaded colors in the figure (from white to bluish). In addition, there is no explanation of the arrows indicated in the figure.

We explained the sketch in the caption (also blue fronts and arrows)

5     This is the reviewer’s personal interest. Although the operation manual of the LabFreeze apparatus may not allow, how about the alternating direction of the freezing fluid silicon oil with a switching bulb for a short period? Is the observed inhomogeneity of the cooling efficiency related to the flow direction of the freezing fluid?

We thank the reviewer for this valuable remark. The LabFreeze system cools through the bottom plate. As the block is quite thin, we assumed homogeneity within the Z plane (which is not completely correct for certain). Through our analysis also now performed via PCA we saw no differences between the samples very close to the long sides of the LabFreeze (Samples 6,4 and 1, you may compare Figure 1 and 5). Therefore, within the LabFreeze we think that alternating the pumping will not help much. However, we believe that the Huber thermostat is capable of switching pumping direction of the fluid. Therefore, this may help with complex geometries at larger scale systems (like the PilotFreeze), which is cooled through the cylindrical mantle and in situ through cooling coils. We will certainly propose the reviewer’s idea to our industrial partner for testing!